# Multi-Modal Spectroscopic Assessment of Skin Hydration

**DOI:** 10.3390/s24051419

**Published:** 2024-02-22

**Authors:** Iman M. Gidado, Ifeabunike I. Nwokoye, Iasonas F. Triantis, Meha Qassem, Panicos A. Kyriacou

**Affiliations:** Research Centre for Biomedical Engineering, University of London, London EC1V 0HB, UK; iman.gidado@city.ac.uk (I.M.G.); ifeabunike.nwokoye@city.ac.uk (I.I.N.); iasonas.triantis.1@city.ac.uk (I.F.T.); meha.qassem@city.ac.uk (M.Q.)

**Keywords:** skin hydration, multi-modal, biosensors, wearables, NIRS, optical, bioimpedance

## Abstract

Human skin acts as a protective barrier, preserving bodily functions and regulating water loss. Disruption to the skin barrier can lead to skin conditions and diseases, emphasizing the need for skin hydration monitoring. The gold-standard sensing method for assessing skin hydration is the Corneometer, monitoring the skin’s electrical properties. It relies on measuring capacitance and has the advantage of precisely detecting a wide range of hydration levels within the skin’s superficial layer. However, measurement errors due to its front end requiring contact with the skin, combined with the bipolar configuration of the electrodes used and discrepancies due to variations in various interfering analytes, often result in significant inaccuracy and a need to perform measurements under controlled conditions. To overcome these issues, we explore the merits of a different approach to sensing electrical properties, namely, a tetrapolar bioimpedance sensing approach, with the merits of a novel optical sensing modality. Tetrapolar bioimpedance allows for the elimination of bipolar measurement errors, and optical spectroscopy allows for the identification of skin water absorption peaks at wavelengths of 970 nm and 1450 nm. Employing both electrical and optical sensing modalities through a multimodal approach enhances skin hydration measurement sensitivity and validity. This layered approach may be particularly beneficial for minimising errors, providing a more robust and comprehensive tool for skin hydration assessment. An ex vivo desorption experiment was carried out on fresh porcine skin, and an in vivo indicative case study was conducted utilising the developed optical and bioimpedance sensing devices. Expected outcomes were expressed from both techniques, with an increase in the output of the optical sensor voltage and a decrease in bioimpedance as skin hydration decreased. MLR models were employed, and the results presented strong correlations (R-squared = 0.996 and *p*-value = 6.45 × 10^−21^), with an enhanced outcome for hydration parameters when both modalities were combined as opposed to independently, highlighting the advantage of the multimodal sensing approach for skin hydration assessment.

## 1. Introduction

The epidermis, the top layer of the skin, provides the function of regulating the rate of transepidermal water loss (TEWL) through the skin barrier [1]. Sebaceous glands secrete oils, forming a lipid layer that reduces the rate of excess TEWL [2]. A damaged skin barrier, e.g., due to eczema, can result in conditions including skin dryness or itchiness, which may be avoided through the use of additional hydration or topical treatments [3,4]. Optical and electrical sensing methods can be employed to accurately measure the hydration of the skin, with each having their own advantages that can be explored to complement one another.

The typical techniques used in measuring the hydration of the skin and its barrier function are electrical-based sensors, with the gold standard device being the corneometer. However, although such devices are widely used and considered accurate, they may exhibit a range of errors that can introduce significant measurement discrepancies which have not been adequately addressed in commercial devices [5]. The major errors can be grouped into two main categories: (i) errors that relate to the electrode–skin contact properties, and (ii) errors due to the nature of the measurand, given that the corneometer does not directly measure water content [6,7,8,9,10].

Errors in category (i) include variability as a function of contact area and geometry, electrode material, and applied pressure [5]. These errors can be associated with the fact that such instruments feature a two-electrode (bipolar) configuration. The corneometer’s principle of operation is described by the wider theory of bioimpedance measurements. Bioelectrical impedance (or bioimpedance) sensing is applied in a broad range of biomedical applications, including body composition estimation, cardiac output measurements, cell culture monitoring, and monitoring of skin hydration, to name a few [9]. The electrical properties of biological tissues are influenced by factors including their morphology, physiology, and pathological conditions [8,9].

The primary source of errors in bipolar bioimpedance systems, including typical corneometers, relates to the electrode–electrolyte double layer introducing a contact impedance that can significantly interfere with the measured value [10]. As measurements are taken across a range of frequencies (typically up to 1 MHz), the detrimental effect of the contact impedance varies significantly, especially when small electrodes and/or low frequencies are employed [11,12].

A commonly employed method for mitigating the impact of contact impedance on measurements involves utilizing four electrodes [12,13,14,15]. Referred to as tetrapolar electrical impedance measurement (TEIM), this technique effectively reduces the adverse effects of the double layer on the measurement. The appropriateness of the tetrapolar approach for measuring hydration has been demonstrated in the literature [6,16]. It is, therefore, desirable to compare the effectiveness of tetrapolar versus bipolar hydration measurements to assess the significance of the double layer errors in electrical sensing methods.

Bioimpedance hydration sensing relates to the skin’s electrical properties, like permittivity and conductivity, whose combined values vary with different water content levels. As the measurements reflect water content only indirectly, they are expressed in arbitrary values [6]. This reliance on electrical properties results in errors in category (ii), including decreased sensitivity at high hydration levels, susceptibility to environmental variations, and fluctuations with electrolyte concentrations. These errors can be minimised by sensing water content directly, which can be made possible through optical sensing based on the capability to directly detect water-related bands present within the spectral response [6,17].

Recently, there has been a surge in the research on optical sensing methods aimed at obtaining parameters for skin properties [6]. The presence of specific features within the absorption spectra of human skin can allow for a more detailed analysis of the skin’s constituents. Although optical sensors reach smaller depths than their electrical counterparts, they can still penetrate beyond the epidermis and provide optical information on specific skin properties [6,18]. The absorption spectra of water are recognized to have certain peaks present at particular wavelengths, the most documented being 970 nm and 1450 nm, with their intensities associated with the amount of water present within the cells. Thus, the sensitivity of recording water presence is increased at these set wavelengths and has been observed to display a positive correlation to absorbance levels. Optical methods are sensitive to OH and HOH bands; hence, it can be argued that this is a more direct measurement of water content. However, this is not necessarily hydration, which is based on multiple aspects. This presents the need for techniques to focus on different aspects of skin to provide a more holistic approach [19,20].

Here, we will investigate both of the main sensing modalities that can potentially eliminate the aforementioned common corneometer errors (i) and (ii). The multimodal sensing approach, i.e., the use of both electrical and optical sensing modalities, can allow for increased sensitivity and validity in bodily measurements by providing a multi-layer approach to assessment. Combining both these techniques can pave the way for the next-generation skin hydration assessment devices [6,21,22].

This paper outlines the ex vivo and proof-of-concept in vivo testing of custom-designed optical and electrical-based sensing devices, assessing their precision and reliability individually and evaluating their combined impact. The ex vivo investigation involved employing the desorption of porcine skin over a 6 h duration at room temperature, with measurements taken at fixed intervals. Furthermore, an in vivo indicative study was conducted to ascertain the sensors’ accuracy when applied to human skin by examining their raw outputs, considering various affecting parameters. Analysis of both methodologies facilitated comprehensive insights into the efficacy and advantages of employing multi-modal approaches for skin hydration measurement. The comparative assessment of the individual techniques highlights their potential complementarity in enhancing the precision of skin hydration measurement.

## 2. Materials and Methods

### 2.1. Design and Development of the Optical Sensor

A multi-wavelength optical sensor was designed and developed, comprising 4 LED wavelengths and a single central photodiode. The photodiode employed in this experimental configuration was the LAPD-09-17-LCC, a photodetector with a large surface area based on InGaAs (indium gallium arsenide) semiconductor technology. InGaAs sensors are known to have high sensitivity and responsiveness within the wavelength range spanning 900 nm to 1700 nm, appropriate for the LEDs selected [21].

The selected light-emitting diodes (LEDs) were within the near-infrared wavelength range, i.e., 975 nm, 1050 nm, 1300 nm, and 1450 nm, each equipped with a dome lens. Among these, the 1050 nm LED was characterized as the reference wavelength due to its minimal interference from water absorption bands at that specific wavelength [22].

A 3D-printed casing enclosure was created to surround the printed circuit board (PCB) comprising the optical sensor in order to provide a consistent spacing for an optical window whilst permitting minimization of the occlusion effects due to direct skin contact. The optical window utilized was a Raman-grade calcium fluoride (CaF2) window, permitting near-infrared wavelengths to pass through without significant absorption [23]. Watch straps were attached to the PCB casing to allow the sensor to be used as a wearable device on the wrist. The final wearable optical sensor and the 3D-printed casing can be seen in Figure 1 and Figure 2, respectively.

The developed optical sensor was interfaced with a processing unit named ZenPPG [24]. This ZenPPG system encompasses essential components for producing the output signals, including a transimpedance amplifier circuit for converting the photodiode’s output current into a corresponding voltage signal, current sources responsible for supplying power to the LEDs in a multiplexed manner, and a composite board structure housing a microcontroller, along with power distribution components. The output of the ZenPPG processing system was connected to a computer via a data acquisition card (DAQ) (NI USB-6212), facilitating data pre-processing and a graphical representation of the results using LabVIEW software (version 2021 SP1 F1). Within the LabVIEW Virtual Instrument (VI) block diagram, a DAQ assistant function was used to consolidate the output signals emanating from the four LEDs. The maximum DC voltage values were extracted from these raw signals and stored in textual formats for further analysis in MatLab (R2023a, 9.14.0.2206163).

### 2.2. Design and Development of the Bioimpedance Sensor

The block diagram of the bioimpedance magnitude measurement system used in this experiment is shown in Figure 3. It consists of an AC current generation stage connected to a pair of current-injecting electrodes and an impedance magnitude measurement stage connected to a separate pair of voltage-measuring electrodes. The current injection comprises a Howland current source (U1). The measurement stage consists of the blocks U2-U4, comprised of an instrumentation amplifier, an active full-wave precision rectifier, and an active second-order low-pass filter. The fabricated system used in the experimental work is shown in Figure 4.

The component U1 is a mirrored-enhanced Howland current source (MEHCS), allowing for a fully floating load and featuring a higher output impedance relative to other voltage-controlled current source architectures. The output current flowing through the load impedance |Z| of the biological sample under test is
(1)$$I_{inj}=g_{m}V_{ref}$$
where $g_{m}$ is the transconductance of the current source (here: 1 mS). U2 is the instrumentation amplifier measuring the resulting voltage across the porcine skin.

The instrumentation was connected to a commercially available [25] tetrapolar conductivity sensor featuring electrodes *E*_1–4_, shown in Figure 5, specified for a frequency range of 100 Hz to 10 kHz.

The current injection electrodes (*E*_1,4_) were spaced 3 mm apart, whereas the voltage measurement electrodes (*E*_2,3_) were positioned with a 2 mm separation. Theoretically, the penetration depth is approximately equal to half the inter-electrode distance [26]; thus, the approximate sensing depth was 1.5 mm. Given that the average skin thickness is less than 6 mm [27,28,29], this sensor predominantly covered the top layers of the skin. The impedance of the porcine skin measured by the electrodes could be modelled using a simplified lumped-impedance network of capacitors and resistors, as shown in the diagrammatic representation in Figure 6, where C1 and R1 represent the interface impedance between the skin and electrodes. The bioelectric field of the injecting electrodes is also shown, traversing the target skin impedance |Z| [30]. The labelled setup for the bioimpedance system during the ex vivo experiment can be seen in Figure 7.

### 2.3. Ex Vivo Experiment on Porcine Skin to Compare Developed Optical and Bioimpedance Sensors

A 6 h ex vivo experiment was conducted using porcine skin to assess the measurement of water content from the developed optical devices during a desorption test. A sample of porcine skin was secured and cut to remove any extraneous adipose or connective tissue. The skin sample had a thickness of 3 mm, consisting of the skin layer and a thin layer of fat. In order to establish optimal hydration, the sample was placed in a humidity chamber set at a controlled relative humidity (RH) of 90%, with the temperature maintained at 25 °C over a duration of 12 h. Porcine skin is a suitable model for testing skin hydration measurement devices due to its high similarity to human skin in terms of optical properties and tissue layer composition. It offers reproducibility and consistency, making it appropriate for use in controlled ex vivo experiments [31].

Once the sample had achieved its augmented hydration level, it was placed directly onto an analytical scale, where its initial gravimetric measurement was recorded. Positioned on the left side of the sample was the developed optical sensor, connected to its external ZenPPG system [24] and DAQ card (National Instruments, Austin, TX, USA). In the centre of the sample, a reflectance fibre optic probe was suspended 1 mm from the skin surface and connected to a commercial benchtop spectrophotometer. On the right side of the sample was the bioimpedance sensing device, held in place with a small weight to prevent movement of the lightweight sensor.

In terms of measuring the impedance (Z), it can be represented by a simplified lumped impedance representation of the human skin, shown in Figure 8 [14], where Re and Ce represent the epidermis and Rd represents the dermis and underlying subcutaneous tissues. From this perspective, porcine skin is a relatively accurate representation of human skin [32]. Both bipolar and tetrapolar electrode configurations were used in the experimental procedure. The tetrapolar measurements were minimally unaffected by the presence of the two R1-C1 impedances, unlike the bipolar measurement. The latter was implemented by shorting the voltage measurement and current injection electrodes to form a single pair by switching S1 and S2 in their on-positions, as shown in Figure 6. For each setup of electrode configuration, an output constant current of 25 µA was generated and injected into the porcine skin.

The initial mass after setup was recorded to be 242.35 g, where the initial readings from the 3 measurement devices were also taken. Although the developed optical sensor measurements were recorded in a continuous manner, the other 3 measurement techniques required recordings to be taken simultaneously in 20 min intervals throughout the 6 h experimental timeline. The overall setup for this experiment can be seen as a flow chart, shown in Figure 9.

### 2.4. In Vivo Experiment to Compare Optical and Electrical Sensors to Corneometer

A small-scale in vivo indicative case study was also conducted to further validate the developed sensors and to confirm their accuracy when used on human skin. This case study is fundamental for several reasons. Firstly, the use of in vivo testing involving human participants provides a more practical applied assessment of the sensors’ performance and validity. While ex vivo studies can offer valuable insights, the application in humans reflects their effectiveness when used in the real world, taking into account the complexities and distinctions of skin and inter-individual differences. These measurements were correlated against the gold-standard corneometer device.

The indicative case study involved mean averaged measurements taken from both forearms of a single participant under two conditions. The first condition involved a dry forearm, serving as a baseline reference. This condition represents the natural state of the skin, with no external applications or influences. It allows for the assessment of the sensors’ ability to capture hydration of the skin in its typical state. The second condition involved a damp forearm with water applied using a soaked pad, emulating a common scenario when the skin is exposed to external elements or is indicated to be well hydrated in terms of internal water intake. This allowed for the sensors’ capacity to accurately respond to changes in hydration levels to be tested. Figure 10 and Figure 11 below display the sensors’ placement for in vivo measurements on human skin, with the bioimpedance sensor in place in a tetrapolar configuration.

Ethical approval for both porcine skin and human forearm measurement experiments was obtained from the City, University of London Research Ethics Committee. Informed consent was obtained from the human participant prior to commencing the study.

## 3. Results—Ex Vivo Experiments

### 3.1. Reference Gravimetric Measurements

Gravimetric measurements are considered the gold standard for the measurement of water content. This method involves determining the change in the weight of a sample before and after the removal of water. The difference in weight is attributed to the water content.

Gravimetric measurements were recorded every 20 min for a total experimental duration of 6 h. These results can be seen in Table 1. There was an extremely noticeable, linear downward trend in weight over time, from 242 g to 192 g, representing the expected decrease in water content as a result of desorption.

### 3.2. Spectral Experimental Results from Optical Sensor

The output from the developed optical sensor generated DC voltage data points over time for each of the four independent wavelengths. Consequently, four distinct time series spanning the six-hour experimental duration were extracted from LabVIEW and visualized using Matlab. To establish a baseline reference wavelength, 1050 nm was chosen due to the minimal presence of distinct water peaks. Subsequently, the signal produced by the 1050 nm LED could be subtracted from the signals originating from the other three wavelengths in later analysis stages. This subtraction process not only negated the need for additional post-processing techniques to correct for external effects, but also enhanced graphical visualization for improved evaluation. The raw data and filtered outputs for the four wavelengths can be seen plotted in Figure 12.

As anticipated, an upward-sloping correlation was evident in the voltage measurements over the total experimental duration. This was consistently observable across all four distinct wavelengths, with the least perceptible variation recorded at 1050 nm, indicating a 0.02 V increment, while the most significant alteration was at 1450 nm, reflecting a pronounced 0.15 V increase, and, thus, an increased rate of change. This empirical observation is consistent with the established literature, which suggests the presence of a distinct and significant water absorption peak at 1450 nm. This results in a heightened sensitivity to alterations in the water concentration within the skin. As the water concentration within the skin decreases, due to the desorption process, the reduced water content results in a diminished attraction for water molecules under incident light at 1450 nm. Consequently, this reduced water concentration exhibits an increase in reflectance values, thereby causing the notable 0.15 V rise which was observed.

The performance of the developed optical sensor was also compared against a conventional spectrophotometer interfaced with a reflectance fibre optic probe. The results provided validation for the reliability of the data obtained with the developed sensor, primarily attributed to the observed linear relationship between the two instruments. This parallelism was established through their concurrent use in a comparable desorption test also employing 975 nm, 1300 nm, and 1450 nm wavelengths. Consequently, this finding significantly enhanced the credibility of the developed sensor as a valid measurement instrument for characterizing alterations in water concentration. The spectrum in Figure 13 shows the changes in the water from the porcine skin as time progressed during the TEWL.

### 3.3. Experimental Results from Bioimpedance Sensor

In the initial 120 min of the experiment, porcine skin bioimpedance measurements from bipolar and tetrapolar electrode configurations were taken, as shown in Figure 14 and Figure 15, respectively, where data recorded at 20 min intervals are indicated by the markers and a spline interpolant was used to generate the data trend traces. Measurements were carried out for frequencies of 1 kHz, 5 kHz, and 10 kHz. The selection of frequencies was influenced by the bandwidth constraints of the conductivity sensor and were lower than the typical maximum bandwidth of corneometers (about 0.9–1.2 MHz), but within range of the lower end of their reported bandwidth [33,34,35], making the chosen values relevant.

As anticipated, in all cases, impedance increased over time, reflecting transepidermal water loss from the surface of the porcine skin. The impedance values at 5 kHz and 10 kHz were lower, which was attributed to the decreased reactance of the tissue’s reactive impedance component at higher frequencies. Notably, the bipolar configuration (denoted as B in the legend) exhibited higher impedance baseline values compared to the tetrapolar measurements (denoted as T), indicating the influence of electrode/electrolyte interface impedance. The impedance variation over the course of the experiments seemed more pronounced in the tetrapolar results.

The overall impedance variation in each case was better illustrated through the plotting of the percentage impedance magnitude increase, shown in Figure 16 and Figure 17 for the bipolar and tetrapolar configurations, respectively. The percentage increase for each data set was calculated using the formula in Equation (2):(2)$$Percentage\;Increase=\frac{Final\;Value-Starting\;Value}{Starting\;Value}\times 100$$

A percentage increase of >21% in impedance magnitude was observed in both cases of electrode configuration. However, the bipolar configuration exhibited a lower percentage increase in impedance at lower frequencies.

In addition, as seen in Figure 18 below, the corneometer was used to assess the hydration status of the porcine skin sample at the start and end of the experiment. These results provide valuable confirmation of the sensor’s accuracy and reliability in assessing changes in skin hydration.

## 4. Results—In Vivo Experiments

The in vivo indicative case study was an investigation aimed at further validating the efficacy and precision of both developed sensors when applied to human skin. The transition to in vivo experimentation represents a fundamental step towards ascertaining the sensors’ practical utility and reliability for real-world applications. Human skin exhibits various complexities and inter-individual differences that can influence measurements, rendering this study an essential link between laboratory and practical usage. To establish a robust benchmark, the data obtained from the developed sensors were correlated against the gold-standard corneometer device. Two conditions were examined on the forearms of the participants: dry skin (baseline) and damp skin (simulating external influences and changes in water content).

Table 2 displays the results obtained from in vivo experiments, including measurements obtained using the bioimpedance sensing device, optical sensor, and corneometer, under both dry and damped skin conditions. Bioimpedance measurements were performed at 1 kHz, while optical sensor measurements were also taken at the wavelength of 1450 nm. Through Kruskal–Wallis testing, it was found that the *p*-value was 0.0073; thus, we are able to reject the null hypothesis and assume statistical significance with these results.

The introduction of ionized water on the skin resulted in a reduction in the measured impedance magnitude. This can be attributed to the substantial influence of dissolved ions on the electrical conductivity of water, providing it with the capacity to conduct electricity. In the skin impedance measurements, the incorporation of ionic water augmented conductivity by enhancing the mobility of ions. Consequently, this augmentation led to a decrease in impedance levels at the measurement point. The anticipated decrease in impedance magnitude when applying ionized water to the skin aligned with corneometer measurements, indicating an increase in hydration levels following water application. Trends observed from the optical sensor output presented a decline in measured voltage with increased hydration, analogous to the bioimpedance sensor.

Figure 19 presents a bar chart illustrating measurements from the three systems detailed in Table 2. Notably, under dry skin conditions, optical measurements exhibited higher reflected voltage compared to damp skin, indicating reduced light absorption. Bioimpedance measurements displayed a higher impedance value for dry skin than for damp skin, which was attributed to decreased ionic presence on the skin surface due to reduced water content. Corneometer values were higher for damp skin than dry skin, as anticipated due to increased capacitance resulting from elevated water levels on the skin surface. While all systems and measurements demonstrated agreement, the corneometer exhibited a higher rate of change between the two states, from dry to damp skin, with the optical technique being the least reactive.

## 5. Statistical Analysis of Results

### Comparative Analysis of Results Using Regression Techniques

Multiple linear regression (MLR) was selected as the modelling method for understanding hydration dynamics through various measurement variables due to its ability to handle multiple independent variables influencing a single dependent variable. This method provides interpretable coefficients, allowing for a quantification of the impact of each variable on hydration status, which is crucial for understanding the specific contributions of different measurements. The assumption of linearity aligns with the expectation that the relationships between measurements and hydration are reasonably linear. Additionally, the statistical significance tests offered by multiple linear regression aid in identifying which measurements significantly contribute to predicting the dermal water content [36].

The results from the MLR model involved three independent variables (x_1_, x_2_, and x_3_), being the optical, bioimpedance, and gravimetric measurements, predicting a dependent variable (y). The model equation is expressed as (y = β_0_ + β_1_x_1_ + β_2_x_2_ + β_3_x_3_), where β_0_ is the intercept and β_1_, β_2_, and β_3_ are the coefficients for the respective variables [36]. The estimated coefficients indicate the contribution of each independent variable to the dependent variable. The intercept (β_0_) was estimated to be −3657.4, representing the expected value of y when all independent variables are zero. The coefficients for x_1_, x_2_, and x_3_ were 1411.8, −0.49935, and −0.94791, respectively. These coefficients represent the expected change in y for a one-unit change in the independent variable, holding other variables constant. The statistical significance of each coefficient was assessed through t-statistics and associated *p*-values. The t-statistics measure the number of standard deviations away from zero that a coefficient is located. The lower *p*-values obtained, 2.9697 × 10^−10^, 0.00066671, 6.594 × 10^−11^, indicate that each variable was statistically significant in predicting y.

The model fit was assessed through various metrics. The R-squared value of 0.996 suggested a very strong fit, while the adjusted R-squared value remained high at 0.995, supporting the model’s reliability. The root mean squared error (RMSE) was 0.501, representing the average prediction error. The R-squared value of 0.996 was very high, which can be justified due to the experiments being carried out ex vivo on porcine skin. Typically, there would be an expected decline in this correlation on human skin due to other affecting factors, e.g., skin texture and composition.

The F-statistic tested the overall significance of the regression model. The low *p*-value (6.45 × 10^−21^) suggests that at least one variable was significant in predicting y, affirming the overall model’s significance. The overall results from this MLR model can be seen in Table 3. Figure 20 displays the actual vs. predicted plot, which visualizes the model’s predictive performance by plotting the actual gravimetric measurements against the predicted values. The R-squared trend over time visualizes how the R-squared changed over time. This provides insights into how the model’s performance varied across different periods, as it was indicated to stay consistently at its high significance level of 0.996. The bar plot for *p*-values visually compares the *p*-values associated with each variable, being optical, bioimpedance, and gravimetric measurements. This helps to identify which variables had a more significant impact on the dependent variable, which was demonstrated to be the output obtained from the bioimpedance sensor, exhibiting a significantly small *p*-value. The results can be seen below in Table 3, and output plots are displayed in Figure 20.

Figure 21 displays the magnitude of the voltage change (V) for both the optical and bioimpedance results in response to each unit in gravimetric decline. This shows a greater magnitude of voltage change per gravimetric unit in the bioimpedance over the optical measurements. Figure 22 presents the actual vs. predicted plots for the optical and bioimpedance output alone against the predicted gravimetric TEWL response. The MLR models were run for each technique, with the R-squared values of 0.888 and 0.97 and *p*-values of 1.66 × 10^−9^ and 2.09 × 10^−4^, respectively, expressing strong correlations for both individually, but more so in the electrical results. In addition, Figure 23 displays the box plot outputs for the Kruskal–Wallis statistical analyses for the three measurement techniques, presenting the spread and variance of each technique.

## 6. Discussion and Conclusions

Optical and electrical sensing instrumentation and methods were designed and developed for the measurement of skin hydration, which was tested through an ex vivo experiment on porcine skin. This paper detailed the construction of these sensing devices and demonstrated their measurement accuracy and validity when compared to multiple standard reference devices. The merits presented through the results from both sensors have aided in the proposition that the combination of these techniques into a multi-modal sensing method would be extremely beneficial, as it would further enforce the precision of skin hydration measurement.

The conducted ex vivo desorption study was undertaken on porcine skin to systematically assess the efficacy and measurement capabilities of the two developed sensors in response to water desorption from the skin sample, signified by the gravimetric measurements. In addition to the comparative analysis between the two developed sensors, the optical device’s performance was compared against a conventional spectrophotometer. Simultaneously, the bioimpedance sensor was benchmarked against the gold standard corneometer device, ensuring the validation of its readings.

The outcomes derived from the developed optical sensor were graphically represented over the course of the experiment, presenting a positively increasing trend. The initial phase of the optical experiment exhibited a steep incline in voltage readings, attributable to greater water concentrations within the cells. Subsequently, a more gradual and sustained increase in voltage was observed through the remainder of the experiment. The voltage values were assumed to be directly correlated with the reflectance values detected by the photodiode, as both parameters exhibited an inverse relationship with the absorbance. This emphasizes the effectiveness of the newly developed optical sensor in precisely tracking alterations in water through the skin. These findings were confirmed with results acquired concurrently from a spectrophotometer, wherein the observed alterations over time mirrored those identified with the developed optical sensor.

These optical findings underscore the sensitivity of optical skin hydration measurements to changes in water concentration, with pronounced variations at wavelengths closer to water absorption peaks, such as 1450 nm, and more moderate, yet significant, changes at wavelengths further from these sharp peaks, such as 975 nm and 1300 nm.

For the bioimpedance collection of results, both tetrapolar and bipolar connection setups were utilized in this experiment. Tetrapolar setups are favoured for their accuracy, as they help to minimize errors associated with electrode–skin contact impedance. This method provides precise measurements of skin impedance, making it well-suited for research and clinical applications where accuracy is paramount. On the other hand, bipolar measurements are simpler and more straightforward, requiring fewer electrodes. However, bipolar measurements are more sensitive to variations in electrode–skin impedance and may be less accurate than tetrapolar measurements, especially with smaller skin areas. Depending on the application, improvements to the bipolar topology can be made by employing a four-electrode/tetrapolar configuration, where one pair of electrodes is used to inject the current while the other one carries out the sensing voltage. In the latter, one pair of electrodes injects the current, while another pair measures the voltage, greatly minimising the effects of the electrode–electrolyte interface and enhancing measurement accuracy [11,12].

The bioimpedance results presented in this research have shown an increase in impedance over time for all frequency data sets, with lower frequencies, i.e., 1 kHz, presenting higher values due to the more pronounced effect of the capacitive properties of biological tissues at low frequencies. Furthermore, there was a percentage increase of >21% in impedance magnitude observed in both cases of electrodes configuration, with the bipolar configuration expressing a lower percentage increase at lower frequencies than the tetrapolar configuration. This effect arises from the predominance of interface impedance in a bipolar electrode configuration as opposed to a tetrapolar configuration. This dominance significantly affects and may reduce the contribution of the targeted impedance area to a greater extent.

Moreover, the electrode–electrolyte interface impedance was sensitive to charge redistribution with changes to the electrode surface chemistry, and this may be one of the main reasons why bipolar data were less uniformly distributed across the experiment duration [37]. Finally, it is worth considering the detriment to power consumption and dynamic range of the higher baseline impedance observed in the bipolar configuration. The tetrapolar configuration exhibited a very similar and less random trend of increasing impedance for all frequency data sets, making it a more reliable method that is less dependent on the electrode properties.

MLR was used to analyse the results obtained from the developed sensors and to assess their statistical significance. In this case, y represents dermal water content, and x_1_ from optical sensor measurements is noticeably linked to the expected markers, with a substantial coefficient and low *p*-value. x_2_, with a strong negative coefficient, suggests an inverse association with hydration to bioimpedance sensor results. x_3_, with a negative coefficient and low *p*-value, is crucial in predicting dermal water content, representing the time-dependent physiological parameter as the reference measure. The correlation coefficient was an R-squared value of 0.996 and a *p*-value of 6.45 × 10^−21^. This value was higher for the combined output as opposed to the MLR results of the independent techniques, enhancing the advantage of a multimodal measurement method. As mentioned, these values are high due to the model being conducted using ex vivo data, with few effects from human skin variations, such as differences in composition. Additionally, Kruskal–Wallis statistical analysis was performed on the experimental results, which gave an output *p*-value of 1.5389 × 10^−11^. This also outputted the variance spread as a box plot of the measurement techniques, which illustrated the magnitude of change for gravimetric as the greatest and optical the smallest. These findings underscore skin hydration’s multifaceted nature, with diverse variables contributing significantly to understanding the different measurement techniques, which can be combined to contribute to a comprehensive measurement system.

An indicative case study was carried out on the developed devices and correlated against the gold standard corneometer device in two forearm conditions (dry and damp). The study aimed at validating the developed sensors for human skin, with the transition from ex vivo to in vivo experimentation being crucial for assessing practical utility and reliability when inter-individual differences are introduced. The results across all wavelengths consistently showed higher voltage readings for drier skin and diminished readings for increased skin moisture. In the electrical measurements with the bioimpedance sensor and corneometer device, the expected decrease in impedance magnitude observed when ionized water was applied to the skin was in agreement with the corneometer measurements, providing further evidence of an elevated level of water content subsequent to the application of water. This alignment suggests a consistent pattern of increased water within the skin, as corroborated by the parallel findings from both bioimpedance and corneometer assessments. The coherent decrease in impedance levels and the concurrent rise in corneometer readings collectively affirm the moisturizing impact of ionized water on the skin, validating its effectiveness in promoting increased skin hydration. Kruskal–Wallis testing for the in vivo results found the *p*-value to be 0.0073 and a wider variance spread for the ex vivo experiment analysis, therefore allowing the null hypothesis to be rejected and statistical significance to be concluded. In addition to this, it is suggested that, based on the electrode geometry and frequencies of measurements, bioimpedance measurements can be taken to analyse even deeper layers of skin tissues beyond the skin surface. Consequently, the disparity in the response patterns between the bioimpedance sensor and optical measurements emphasizes a more complex understanding of skin hydration and the need to harness their complementarity for a more accurate skin hydration measurement. By examining these diverse conditions, this case study has provided insights into the sensors’ ability to differentiate between various levels of skin hydration. This is crucial for assessing the sensors’ potential applications in the real world and the reliability of their measurements.

In conclusion, this study has demonstrated that the potential advantages of both sensors (optical and impedance), when used in combination, could result in a more effective monitoring modality for skin hydration.

## Figures and Tables

**Figure 1 sensors-24-01419-f001:**
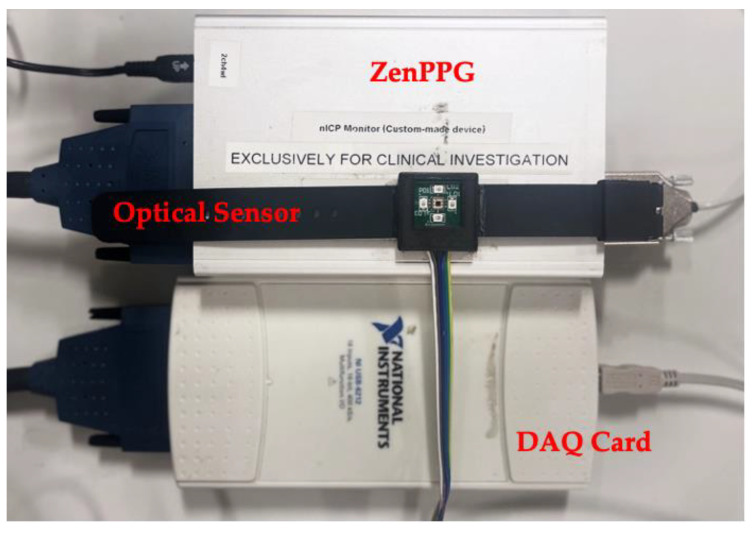
Optical wearable sensor connected to the ZenPPG processing system and the Data Acquisition Card (DAQ).

**Figure 2 sensors-24-01419-f002:**
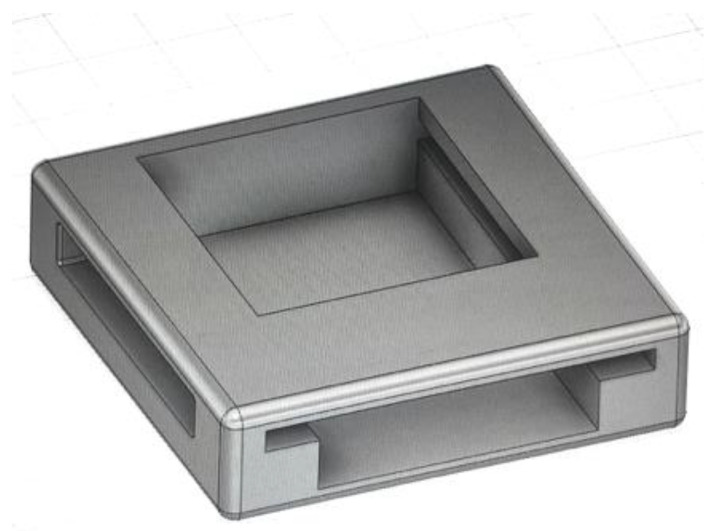
3D-printed casing for optical sensor PCB.

**Figure 3 sensors-24-01419-f003:**
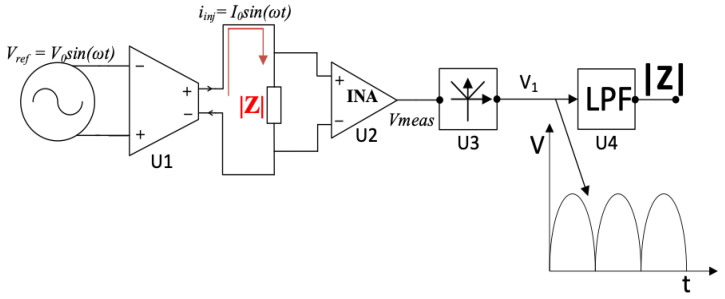
Block diagram of the bioimpedance measurement system.

**Figure 4 sensors-24-01419-f004:**
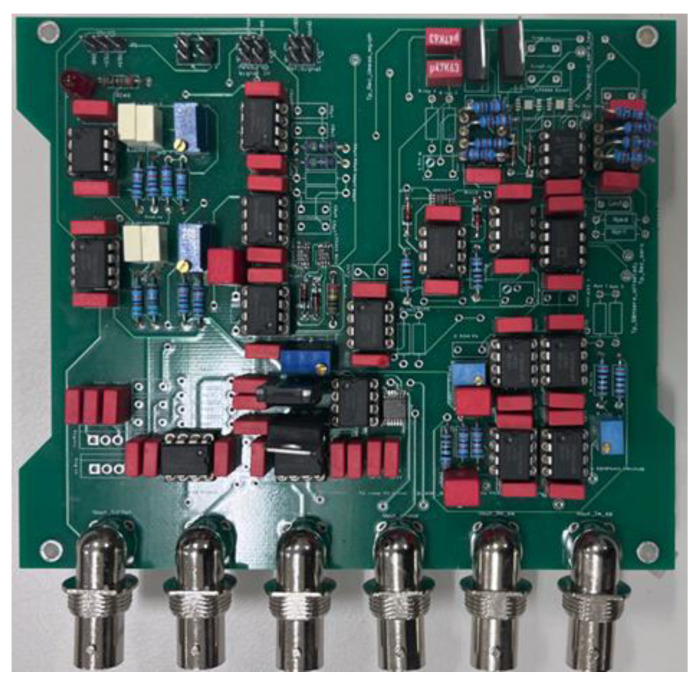
PCB including the developed bioimpedance system.

**Figure 5 sensors-24-01419-f005:**
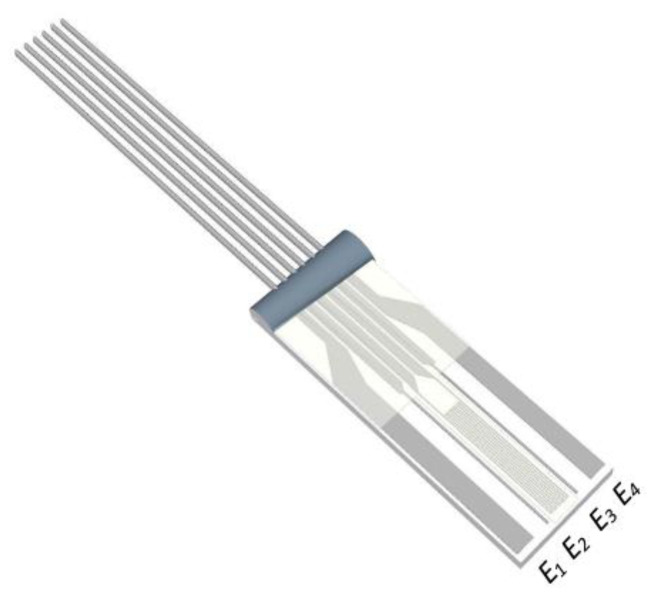
Measuring tetrapolar sensor.

**Figure 6 sensors-24-01419-f006:**
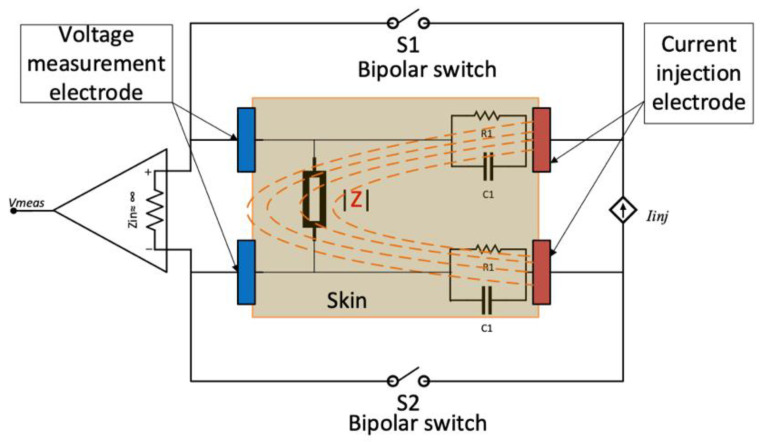
Skin model illustrating bipolar and tetrapolar electrode configuration.

**Figure 7 sensors-24-01419-f007:**
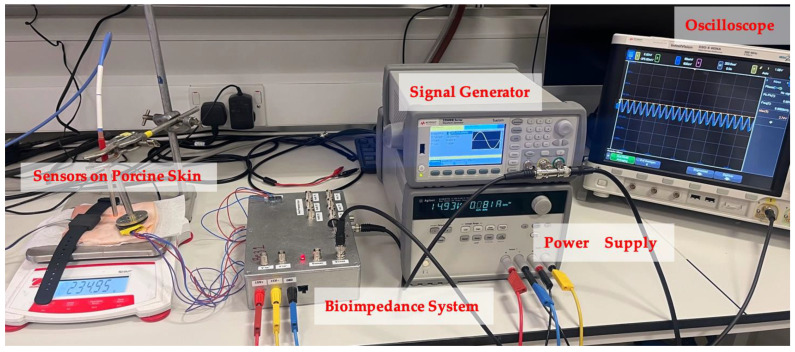
Ex vivo experimental setup for bioimpedance system.

**Figure 8 sensors-24-01419-f008:**
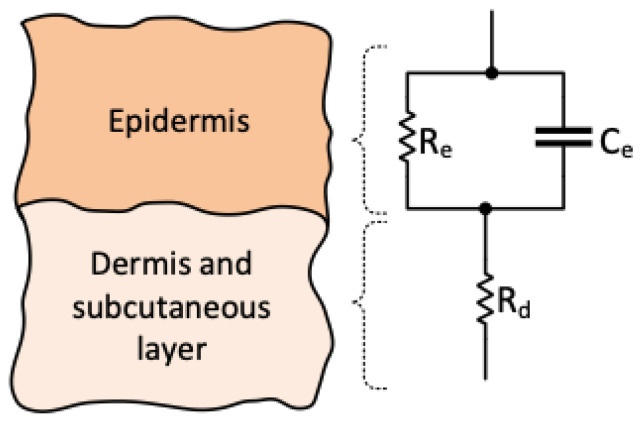
Expanded model of the tissue at the target skin site and its equivalent circuit model.

**Figure 9 sensors-24-01419-f009:**
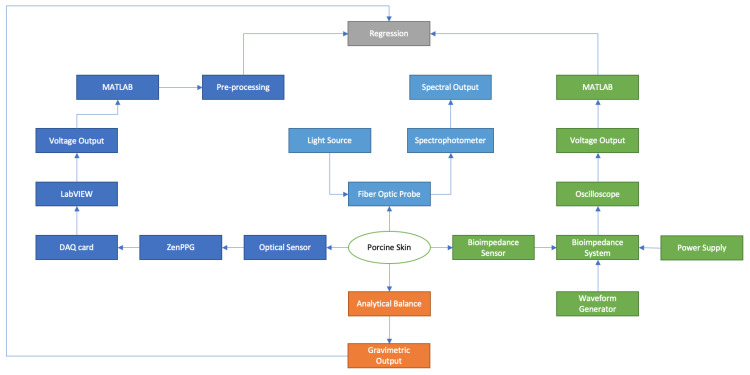
Complete setup and process for ex vivo experiment illustrated as a flow chart (blue = optical, green = bioimpedance, orange = gravimetric measurements).

**Figure 10 sensors-24-01419-f010:**
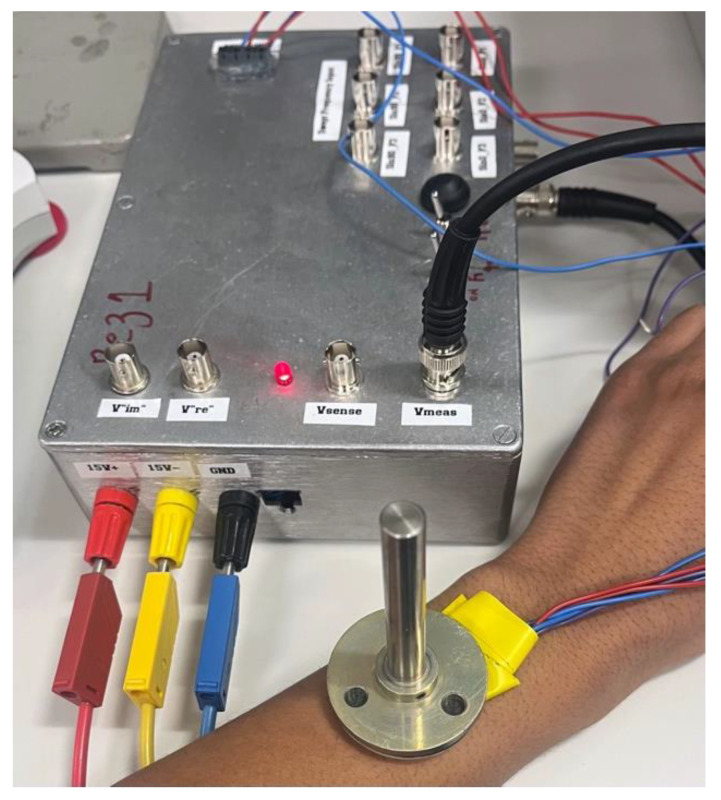
Bioimpedance sensor positioned on human skin with weight to hold electrodes in place.

**Figure 11 sensors-24-01419-f011:**
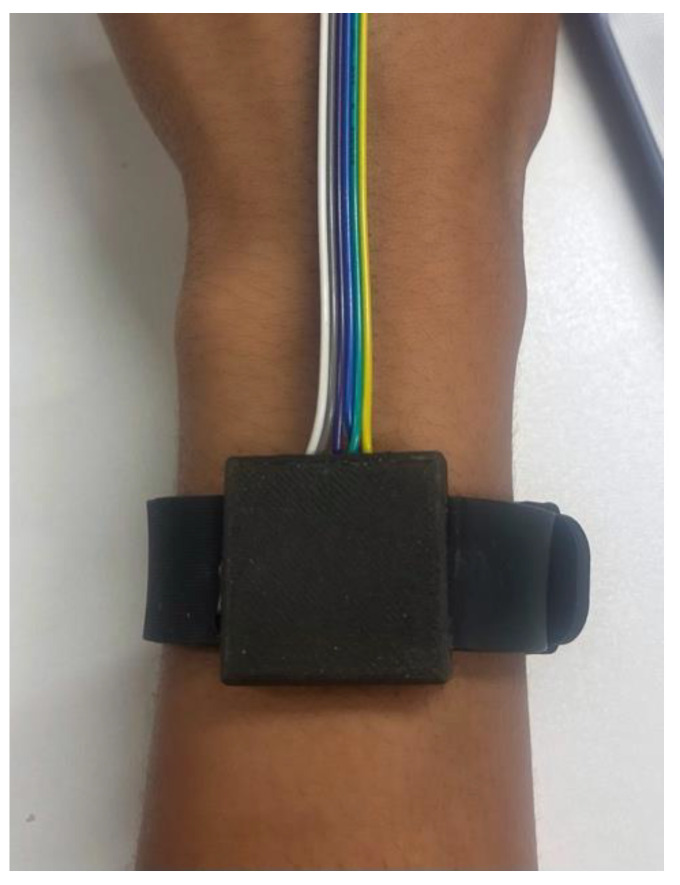
Optical wearable sensor on wrist.

**Figure 12 sensors-24-01419-f012:**
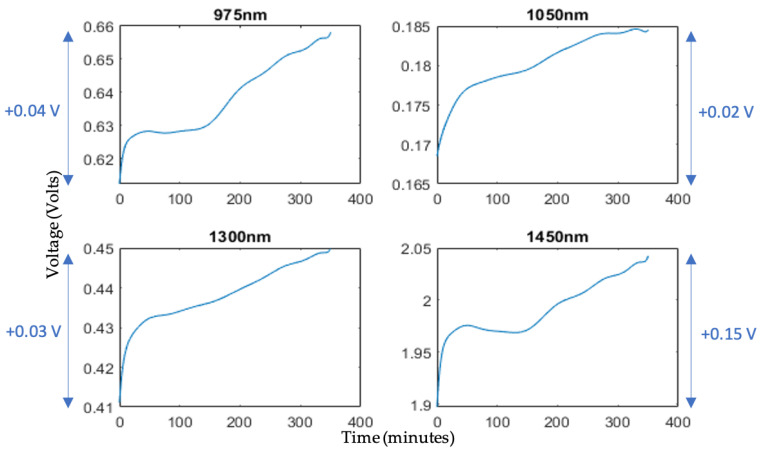
Raw voltage output from optical sensor over 6 h experimental time for selected LED wavelengths, with total incremental increases during ex vivo desorption test.

**Figure 13 sensors-24-01419-f013:**
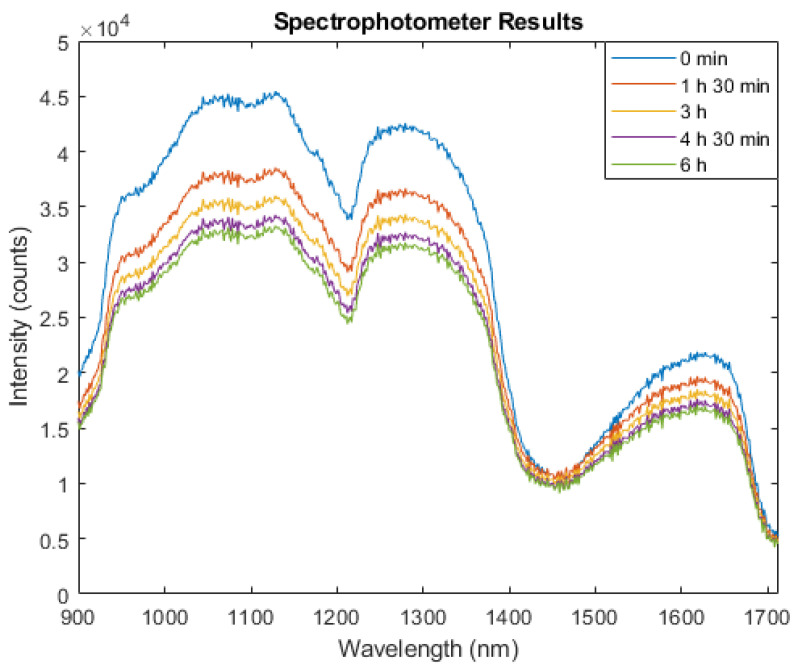
Spectrophotometer output at different timepoints during ex vivo experiment.

**Figure 14 sensors-24-01419-f014:**
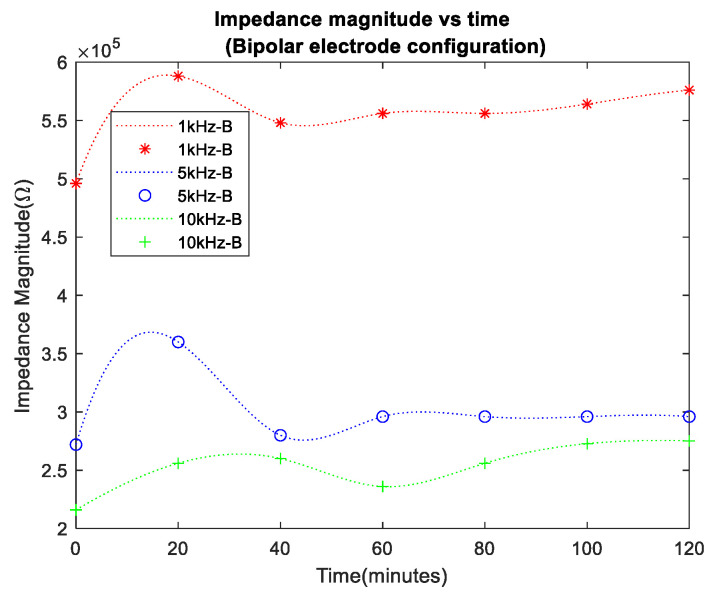
Impedance magnitude vs. time for bipolar electrode configuration.

**Figure 15 sensors-24-01419-f015:**
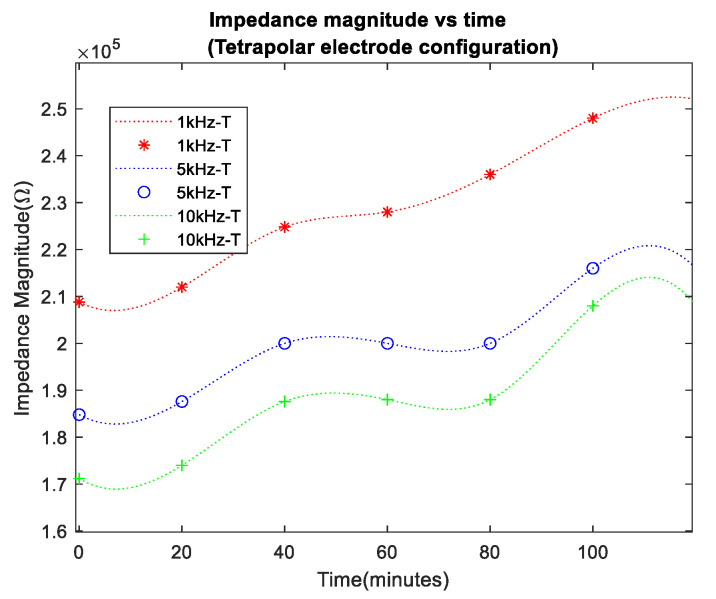
Impedance magnitude vs. time for tetrapolar electrode configuration.

**Figure 16 sensors-24-01419-f016:**
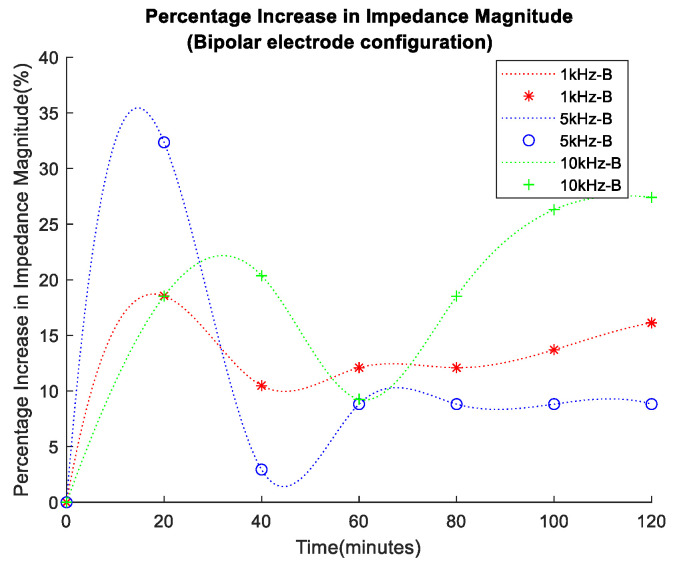
Percentage increase in impedance magnitude for bipolar electrode configuration.

**Figure 17 sensors-24-01419-f017:**
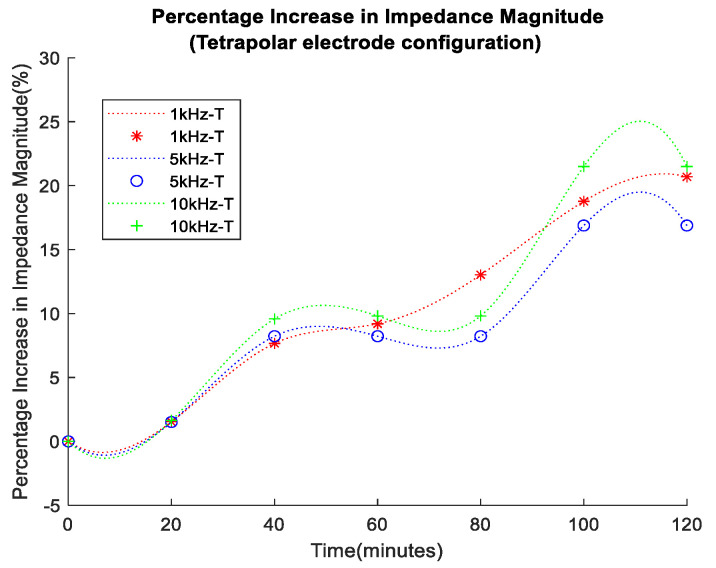
Percentage increase in impedance magnitude for tetrapolar electrode configuration.

**Figure 18 sensors-24-01419-f018:**
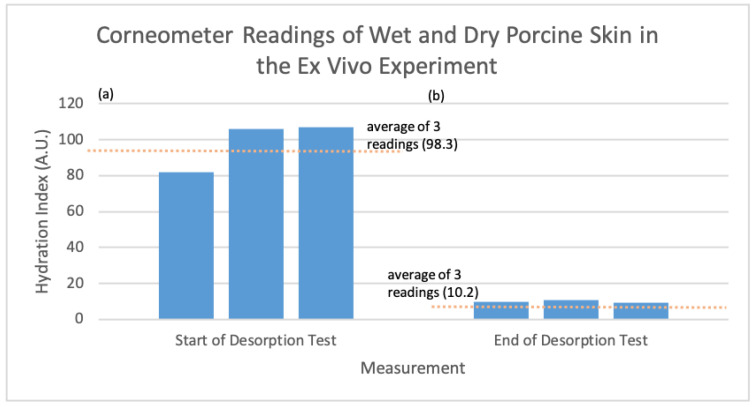
Corneometer output readings for porcine skin sample before and after the ex vivo desorption experiment (Number = Number of Measurements, Corneometer Value = Hydration Index). (**a**) Hydration index of porcine skin before desorption test, (**b**) hydration index of porcine skin after desorption test.

**Figure 19 sensors-24-01419-f019:**
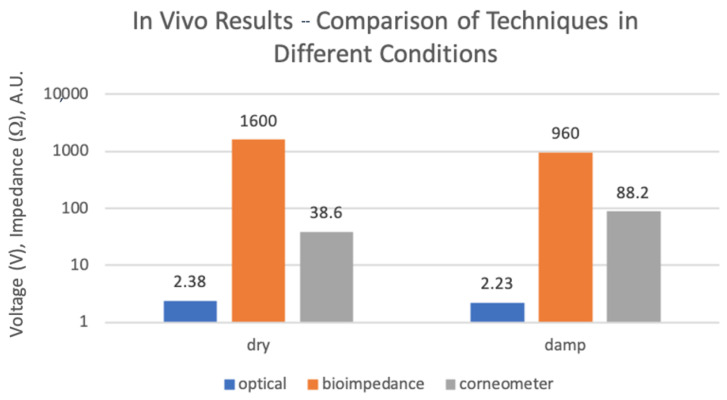
Comparison of outputs from all techniques for in vivo experiment over both conditions (logarithmic scale).

**Figure 20 sensors-24-01419-f020:**
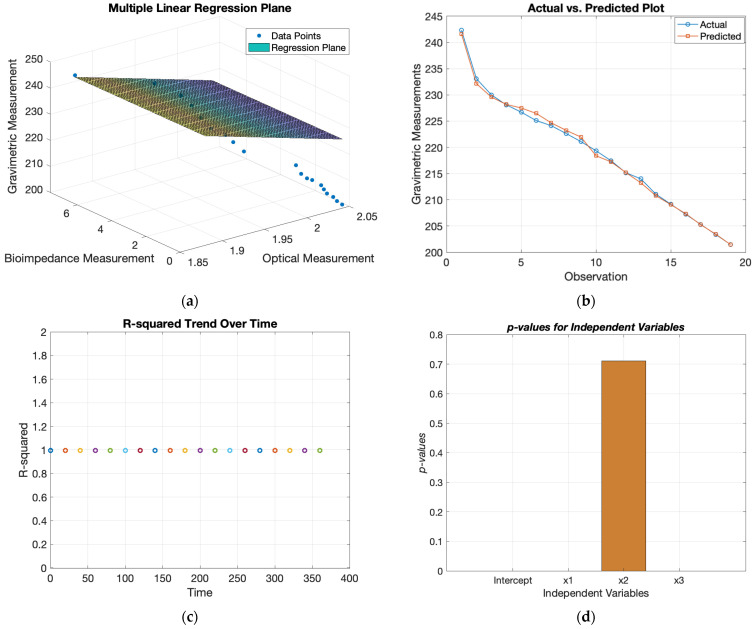
MLR model’s graphical results for optical sensor and bioimpedance sensor voltage outputs against gravimetric measurements for ex vivo experiment: (**a**) plane based on regression coefficients with gradient-representing colours, (**b**) actual vs predicted plot, (**c**) R-squared trend over time, (**d**) *p*-values for independent variables.

**Figure 21 sensors-24-01419-f021:**
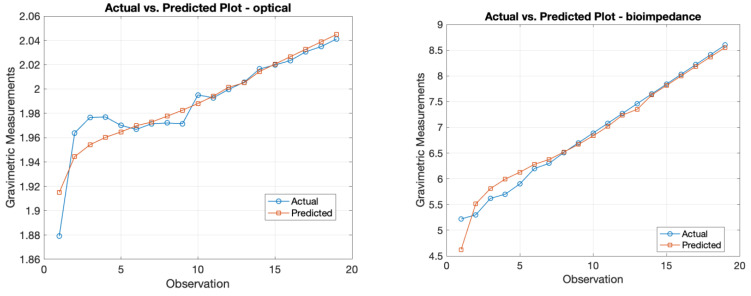
Actual vs. predicted plots of MLR models for optical and bioimpedance results against gravimetric measurements.

**Figure 22 sensors-24-01419-f022:**
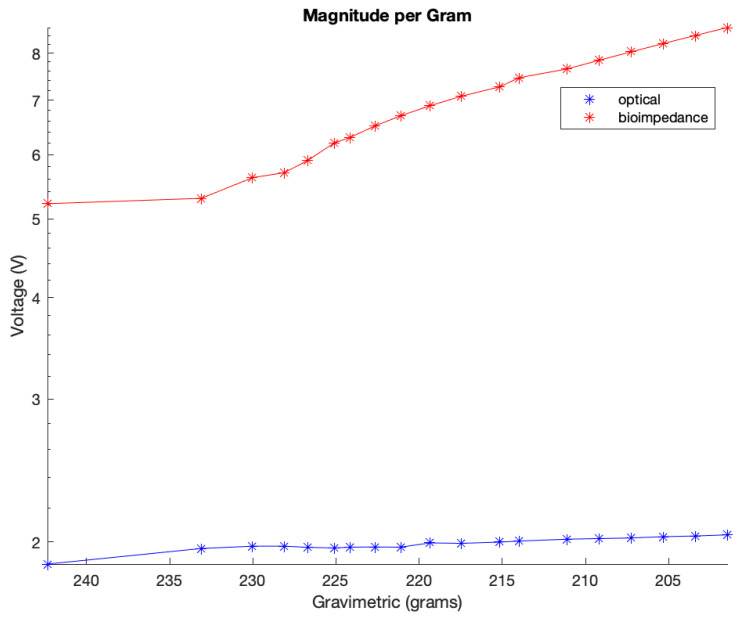
Magnitude of voltage change per gravimetric unit change.

**Figure 23 sensors-24-01419-f023:**
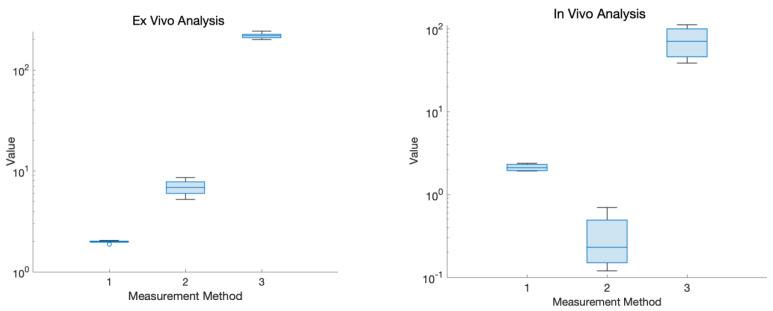
Box plot variance outputs for Kruskal–Wallis statistical analysis for ‘measurement method 1. optical, 2. bioimpedance, and 3. reference measurements’ in ex vivo (**left**) and in vivo (**right**) experiments.

**Table 1 sensors-24-01419-t001:** Recorded gravimetric measurements during desorption test of the ex vivo experiment.

Time (min)	0	20	40	60	80	100	120	140	160	180	200	220	240	330	360
Gravimetric Measurement (grams)	242	233	230	228	226	225	224	222	221	219	217	215	214	193	192

**Table 2 sensors-24-01419-t002:** Bioimpedance sensor results for in vivo experiment over 2 conditions on the forearm.

Condition	Dry	Damp
Optical @1450 nm	2.38 V	2.23 V
Bioimpedance @1 kHz	1.6 kΩ	960 Ω
Corneometer (A.U)	38.6	88.2

**Table 3 sensors-24-01419-t003:** MLR model results for optical sensor and bioimpedance sensor voltage outputs against gravimetric measurements.

	Estimated Coefficient	Standard Error	T-Statistic	*p*-Values	Root Mean Squared Error	0.501
β_0_	−3657.4	267.52	−13.671	7.1367 × 10^−10^	R-squared value	0.996
X_1_	1411.8	97	14.554	2.9697 × 10^−10^		
X_2_	−0.49935	0.11685	−4.2733	0.00066671	Adjusted R-squared value	0.995
X_3_	−0.94791	0.05856	−16.187	6.5946 × 10^−11^		

## Data Availability

The data presented in this study are available on request from the corresponding author.

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
