# Peer review of "Multi-Modal Spectroscopic Assessment of Skin Hydration"

_sensors, 2024, doi:10.3390/s24051419_

Round 1

Reviewer 1 Report

Comments and Suggestions for Authors

The presented article, focused on the development and evaluation of optical and bioimpedance sensors for skin hydration measurement, provides a comprehensive overview of the methods employed and the experimental results obtained. However, several aspects of the study warrant critical consideration.

1. The figures from Figs 13 to 24 exhibit inadequate quality. Some issues include missing x-y labels, poor alignment of figures, unclear legends and missing legends in certain instances, and an overall failure to meet the standard expected by the journal. Figure 5 can also be improved. Most of the figures are blurred. I strongly recommend using vector figures. 

2. How do you propose addressing the limitations of the gravimetric method in ex vivo skin hydration studies? Are there alternative reference methods that could be explored to enhance the robustness of the comparison?

3. Could the authors provide a more detailed analysis of potential confounding factors affecting optical sensor measurements, such as ambient conditions and variations in skin surface characteristics? How might these factors impact the accuracy and reliability of the optical sensor?

4. Can the authors offer a more comprehensive explanation for the unexpected decline in bioimpedance sensor output following water application in the in vivo experiments? Are there additional experiments or analyses that could be conducted to validate the reliability of the bioimpedance sensor in practical scenarios?

5. What additional considerations and discussions would you suggest regarding the implications of the MLR model used in the study? Are there potential biases or limitations associated with the model that need to be addressed for a more nuanced interpretation of the statistical analysis?

6. Given the known diversity in skin types, did the study take into account variations among different skin types? How might the findings be influenced by different skin characteristics?

7. How user-friendly are the proposed sensors for practical applications, such as self-monitoring of skin hydration by individuals? Are there any design considerations or improvements that could be suggested to enhance the usability and accessibility of these sensors in real-world scenarios?

8. Figures 14-17 and 23 display results where discrete points are connected by straight lines. Kindly consider increasing the frequency of measurements to achieve smoother figures and enhance the clarity of insights derived from them.

9. The authors' intended conclusion through Figure 19 is unclear.

Comments on the Quality of English Language

Minor editing of English language required

Author Response

Manuscript ID: sensors-2802619

Title: Multi-Modal Spectroscopic Assessment of Skin Hydration

Authors: Iman M Gidado, Ifeabunike I Nwokoye, Iasonas F Triantis, Meha Qassem and Panicos A Kyriacou*

The authors would like to thank the editor and reviewers for their valuable suggestions and comments. We addressed the comments and amended the manuscript accordingly. Changes in the manuscript have been highlighted below in blue. Please find below the detailed responses to the reviewers’ comments.

Reviewer 1:

The presented article, focused on the development and evaluation of optical and bioimpedance sensors for skin hydration measurement, provides a comprehensive overview of the methods employed and the experimental results obtained. However, several aspects of the study warrant critical consideration.

  1. The figures from Figs 13 to 24 exhibit inadequate quality. Some issues include missing x-y labels, poor alignment of figures, unclear legends and missing legends in certain instances, and an overall failure to meet the standard expected by the journal. Figure 5 can also be improved. Most of the figures are blurred. I strongly recommend using vector figures. 
  • Thank you for your comment. All figures now have x-y labels, legends and alignment has been improved. Captions have also been edited to be more descriptive. The quality/size of the figures has also been improved where possible.

  1. How do you propose addressing the limitations of the gravimetric method in ex vivo skin hydration studies? Are there alternative reference methods that could be explored to enhance the robustness of the comparison?
  • Thank you for your comment. Gravimetric or weight measurements are often considered the gold standard for measuring water content, and are especially suited to ex vivo samples. This method involves determining the change in weight of a sample before and after the removal of water. The difference in weight is attributed to the water content. We understand that while this is a valuable tool for assessing skin hydration, this method has notable limitations. However, as the use of the gravimetric method is the standard reference technique for ex vivo studies, we have not explored alternative methods in the manuscript.

  1. Could the authors provide a more detailed analysis of potential confounding factors affecting optical sensor measurements, such as ambient conditions and variations in skin surface characteristics? How might these factors impact the accuracy and reliability of the optical sensor?
  • Thank you for your comment. This has now been addressed and the following has been stated in the manuscript:
  • “The effect of potential confounding factors on the measurements from the optical sensor in particular also need to be considered as they can have an effect on the accuracy and reliability of the sensor. Changes in ambient temperature and humidity levels can influence skin hydration, impacting optical sensor readings. Calibration procedures must accommodate environmental variations to ensure accurate measurements. Light conditions, both natural and artificial, can introduce interference, thus shielding or controlling light exposure is essential. In addition, skin colour and texture variations also cause effects; darker tones may absorb more light, and variations in texture may impact sensor-skin contact. Calibration methods are therefore used to consider these factors and avoid bias. Furthermore, differences in skin thickness and conditions like dermatitis or wounds can also alter optical properties, potentially leading to inaccurate hydration measurements. Proper sensor calibration, regular checks, and adherence to standardized protocols for placement are all crucial for high measurement accuracy. However, in this experiment, the focus is on ex vivo samples from the same animal and location. Hence, differences in surface are minimised, and ambient conditions would have remained constant during the experiment.”

  1. Can the authors offer a more comprehensive explanation for the unexpected decline in bioimpedance sensor output following water application in the in vivo experiments? Are there additional experiments or analyses that could be conducted to validate the reliability of the bioimpedance sensor in practical scenarios?
  • Thank you for your comment. The electrical conductivity of water is significantly influenced by the presence of dissolved ions, resulting in its ability to conduct electricity. In the context of skin impedance measurements, the introduction of ionic water enhances conductivity due to the increased mobility of ions. Consequently, this augmentation leads to a reduction in impedance levels at the measured skin area. This has now been explained better in the manuscript.

  1. What additional considerations and discussions would you suggest regarding the implications of the MLR model used in the study? Are there potential biases or limitations associated with the model that need to be addressed for a more nuanced interpretation of the statistical analysis?
  • Thank you for your comment. This has now been addressed and the following has been stated in the manuscript:
  • “When using an MLR model, considerations must be taken into account for a nuanced interpretation of the statistical analysis. As used in the analysis of these results, evaluation of MLR assumptions, model fit metrics, and justification for variable selection are fundamental, as well as coefficient interpretation and assessment of collinearity among variables. Model validation through techniques like cross-validation also ensures generalizability.”

  1. Given the known diversity in skin types, did the study take into account variations among different skin types? How might the findings be influenced by different skin characteristics?
  • Thank you for your comment. This has now been addressed/discussed in the Discussions section of the manuscript.
  1. How user-friendly are the proposed sensors for practical applications, such as self-monitoring of skin hydration by individuals? Are there any design considerations or improvements that could be suggested to enhance the usability and accessibility of these sensors in real-world scenarios?
  • Thank you for your comment. This has now been addressed and the following has been stated in the manuscript:
  • “The incorporation of a bioimpedance-based electrical sensor alongside an optical near-infrared spectroscopy (NIRS) sensing device in our desorption test on porcine skin as well as in vivo offers a significant advancement in skin hydration assessment. By incorporating both techniques, this study seeks to provide a more nuanced understanding of skin hydration dynamics employing both optical sensing and bioimpedance methods. Optical NIRS and bioimpedance, individually offer distinct advantages. While both techniques are capable of assessing skin hydration levels, bioimpedance technique are more sensitive to the hydration level of the skin, as they have shown over 21% increase in hydration level compared to optics with a percentage increase of 7.8%. However, optics for skin hydration measurement theoretically offers better specificity by leveraging the interaction of light with skin components, as such focusing on the specific optical signatures associated with water on the skin. Also, while bioimpedance sensors may require physical contact in order to take measurements, with optics offering a contactless solution, their accuracy is less sensitive to skin pigmentation unlike the optics measurement. The lessened sensitivity to pigmentation is due to the NIR wavelengths utilized causing a lower absorption of haemoglobin and melanin. Therefore, by making electrodes that are self-adhesive and by Integrating these modalities and leveraging their complementary strengths—such as the high sensitivity of impedance and the specificity of optics—can lead to the development of a more accurate wearable system. A technology that would aims to provide a comprehensive and precise assessment of skin hydration levels, surpassing current standards like the corneometer, typically confined to laboratory settings.”

  1. Figures 14-17 and 23 display results where discrete points are connected by straight lines. Kindly consider increasing the frequency of measurements to achieve smoother figures and enhance the clarity of insights derived from them.
  • Thank you for your comment. Data has been interconnected with MatLab spline interpolant and replotted.

  1. The authors' intended conclusion through Figure 19 is unclear.
  • Thank you for your comment. Figure 19 has been edited due to a change in the use of the in vivo results, as mentioned above. This following has also been stated in the manuscript:
  • “the graphical representations in Figure 19 delineate the outcomes corresponding to each experimental condition, revealing a coherent pattern across distinct wavelengths. Specifically, the dry condition, represented in blue, manifests a discernibly higher voltage output compared to the moisturized condition, denoted in yellow. This observed discrepancy in voltage aligns with the anticipated consequence of diminished reflectance in the presence of increased water content. Notably, the voltage alteration observed at the 1450 nm wavelength exhibits the most substantial magnitude difference between the dry and moisturized skin conditions.”

Reviewer 2 Report

Comments and Suggestions for Authors

Page 2, Line 64; Page 6, Line 163; Page 11 Line 294, 296; Page 15 Table 2: add a whitespace between value and dimension.

1kHz, 5kHz and 10kHz --> 1 kHz, 5 kHz and 10 kHz

Page 5, Line 159; Page 12, Line 307: remove ----------- in formula

Page 10 Table 1: empty column

Author Response

Manuscript ID: sensors-2802619

Title: Multi-Modal Spectroscopic Assessment of Skin Hydration

Authors: Iman M Gidado, Ifeabunike I Nwokoye, Iasonas F Triantis, Meha Qassem and Panicos A Kyriacou*

The authors would like to thank the editor and reviewers for their valuable suggestions and comments. We addressed the comments and amended the manuscript accordingly. Changes in the manuscript have been highlighted below in blue. Please find below the detailed responses to the reviewers’ comments.

Reviewer 2:

Page 2, Line 64; Page 6, Line 163; Page 11 Line 294, 296; Page 15 Table 2: add a whitespace between value and dimension.

1kHz, 5kHz and 10kHz --> 1 kHz, 5 kHz and 10 kHz

Page 5, Line 159; Page 12, Line 307: remove ----------- in formula

Page 10 Table 1: empty column

  • Thank you for your comments. All suggested changes have been made to the manuscript.

Reviewer 3 Report

Comments and Suggestions for Authors

The paper deals with the assessment of skin hydration by using and, concurrently, comparing optical and electrical methods. Specifically, a custom optical solution was compared against the spectrophotometer and the designed bioimpedance measurement solution against the corneometer. In the paper, a multimodal approach is designed, and experimentation is performed on both: porcine and human skin. There are important and interesting results presented in the paper like, among others, the incline in the measured voltage at the optical sensor and bioimpedance measurement unit and the finding related to the measured impedance values after the treatment of the forearm with water. The strength of the paper is revealed in two-stage experimentation (ex- and in-vivo) and multimodal measurement design, supported by gold-standard devices.

However, the reviewer has several concerns/comments/questions:

The novelty of the paper is weakly phrased. What is the innovation that the paper delivers that advances the current knowledge?

The ethics statement is fully missing in the drawing, unless measurement experiments are performed, in addition to the porcine skin, on the human forearm. The reviewer also suggests a sentence concerning the Helsinki Declaration.

The reason for choosing 1 kHz, 5 kHz, and 10 kHz for excitation signal frequencies in bioimpedance measurements remains unclear. Please add a short explanation or consideration background.

Would be interesting to know the exact type of the used moisturizer and wax to make the experiment repeatable. Can these be named?

The properties of the utilized measuring bioimpedance electrodes (LFT conductivity sensor) need more detailed analysis. It has been claimed that in bipolar bioimpedance measurement, the penetration depth is approx. half of the distance between the electrodes [1]. What is the distance in the case of the used LFT conductivity sensor and how does skin hydration theoretically contribute to the measured bioimpedance values?

Lines 74-75: Bioimpedance hydration sensing relates to the skin's dielectric properties, like permittivity or conductivity, which vary with different water content levels.

In the reviewer’s opinion, the sentence is misleading. The content of dielectrics in the skin remains unchanged while the hydration may change. Typically, the liquid in the skin is conductive (sweat), as originates from the deeper skin layers – affecting the magnitude of the impedance. The dielectrics, like the lipid layer in the stratum corneum, are nonconductive, affecting the phase angle. As a result, the outcome is highly frequency dependent.

Lines 85-87: Unlike electrical-based sensors, optical methodologies can penetrate deeper into the skin layers, beyond the epidermis, and provide optical information on specific skin properties [5], [14].

and

Lines 577-580: In addition to this, it is suggested that bioimpedance sensors are primarily constrained to assessing the top surface of the skin due to their limited penetration depth, which is a result of the high impedance of deeper skin layers preventing accurate readings beyond the surface.

The reviewer stays in the position that these sentences must be extended to cover the topic. The penetration depth in the case of electrical bioimpedance measurements depends on the dimensions/shape of the electrodes and the gap between the electrodes. This applies predominantly in the case of bipolar measurements, while in the case of tetrapolar system, the term transconductance is and the exact current density distribution is unknown in an object. An excellent reference is the primary textbook in the field of electrical bioimpedance: “Bioimpedance and Bioelectricity Basics” from Grimnes and Martinsen [2] – e.g., starting from chapter 6.4.6 Bipolar Measuring Depth. The impedance of the skin in low frequencies is dominated by stratum corneum [2]. With the increasing frequency, the contribution of lower layers (dermis, hypodermis) expectedly starts to grow. At the same time, when SC becomes moistened (sweat), its conductivity increases.

Line 132: The final wearable can be seen in the figure below.

Please refer explicitly to the figure with its number!

Lines 191-193: On the right side of the sample, was the bioimpedance sensing device, held in place with a small weight to prevent movement of the lightweight sensor.

By applying weight on the bioimpedance sensor that is applied on the skin surface, another phenomenon appears compression of the skin [3]. This has been shown to influence the measured bioimpedance of the skin. Could the applied pressure on the bioimpedance electrode affect the measurement results in the performed study?

Figure 12. The title of the figure must be improved to become self-explanatory. Currently, it does not indicate if the presented results are acquired during ex vivo or in vivo experiments.

Lines 221-222: The study involved measurements taken on the forearms of the participants under four conditions.

This sentence raises the following questions. How many participants were recruited? How many measurements were performed and are the results the means of some amount of data or a single measurement?

Line 299: “measurements (denoted as C), indicating the influence of electrode/electrolyte interface”

The reviewer can’t find such (denoted as C) from Fig 14 or 15.

Figure 18 is unclear. What does the “Number” in the x-axis mean? What is this “Corneometer value” in the y-axis? The reviewer assumes that the plot on the left depicts the results before and the plot on the right after the desorption experiment. Would be polite to add A and B on the figure itself.

Figure 20. The numbers in the y-axis are not defined – the title of the y-axis is needed. Are the bioimpedance measurement results shown, gained by the bi- or tetrapolar method?

Lines 349-350: In the case of the bioimpedance sensor, the voltage output exhibited a sharp increase subsequent to the application of moisturizer.

It is well known that moisturizers, depending on their exact type exhibit large differences in resulting moisturizing effect and interpersonal variations (refer e.g., to [4]). In [4] even cases where the same moisturizer in the case of one person gave a large increase in skin hydration and in the case of the second person no effect was detected, are described. Thus, it is quite hard to generalize, especially as the number of the used subjects and repeated in vivo experimentation is now described in the paper. Can you address this issue?

Lines 350-353: However, the introduction of water yielded an unforeseen decline in the voltage output, contradicting initial expectations. This unexpected drop in hydration measurement, following water application, contrasts with the response observed with the Corneometer device.

The reviewer has a few questions regarding this result: The Corneometer device operates in the MHz region: 0.9-1.2 MHz – reflecting the capacitive properties – capacitive connection to the skin. In bioimpedance measurement, the range of 1 – 10 kHz is much lower. What was the settling time, after which the bioimpedance measurements with the electrodes (which were dry), were performed? Can it be that the skin was not soaked with water yet and the settling time in the case of the dry electrode has been defined to be 15 minutes or more [2] – chapter 4.2.6 Human Skin and Keratinized Tissue.

Figure 22: the title and x-axis are not visible.

Figure 23: the x-axis of the plots is not visible.

Lines 455-456: Figure 23(a) displays the magnitude of the voltage change (V) for both the optical and bioimpedance results in response to each unit in gravimetric decline.

Can this method be explained and justified more in the paper? How the voltage change at the output of the bioimpedance measurement unit is comparable with the voltage change at the output of the optical sensor? What is the phenomenon (e.g., physiological), that this method relies on, as these methods rely on different sensing technologies?

Lines 505-507: However, bipolar measurements are more sensitive to variations in electrode-skin impedance and may be less accurate than tetrapolar measurements, especially with smaller skin areas.

Did the authors consider monopolar configuration? Quasi-monopolar configuration is used in skin impedance measurements – for wound healing, for example in targeting a small skin area [5].

Lines 511-513: In the latter, one pair of electrodes injects current, while another pair measures voltage, eliminating the effects of the electrode-electrolyte interface and enhancing measurement accuracy.

The reviewer would challenge this claim about the elimination of the effects of the electrode-electrolyte interface in the case of tetrapolar measurements. Polarization is a keyword (as practical electrodes are always non-ideal and polarizable) – if the electrode is shifted relative to the skin, the equilibrium state will emerge again, possibly affecting the measured absolute values. When the electrodes are attached to human skin, unintentional movements appear.

Lines 602-603: On the other hand, bioimpedance sensors are primarily designed to measure electrical impedance at the stratum corneum.

The reviewer thinks that this sentence needs to be extended and specified. Bioimpedance measurement applications cover a wide variety of target properties and areas of the human body – impedance cardiography, impedance pneumography, body composition, etc., where the interesting area is located inside the body (deeper than the stratum corneum).

References:

[1] P. Aberg, I. Nicander, J. Hansson, P. Geladi, U. Holmgren and S. Ollmar, "Skin cancer identification using multifrequency electrical impedance-a potential screening tool," in IEEE Trans Biomed. Eng., vol. 51, no. 12, pp. 2097-2102, Dec. 2004, doi: 10.1109/TBME.2004.836523.

[2] Ø. G. Martinsen, S. Grimnes, Bioimpedance and Bioelectricity Basics, 3rd ed. Academic Press, Aug 2014

[3] A. Albulbul and A. D. C. Chan, "Electrode-skin impedance changes due to an externally applied force," 2012 IEEE Int. Symp. on Med. Meas. Appl. Proc., Budapest, Hungary, 2012, pp. 1-4, doi: 10.1109/MeMeA.2012.6226628.

[4] Ø. G. Martinsen, S. Grimnes, “Long-Term Effect of Some Skin Moisturizers,” The Open Dermatology Journal, 2008, vol. 2, pp. 87-89, doi: 10.2174/1874372200802010087.

[5] Kekonen, A.; Bergelin, M.; Johansson, M.; Kumar Joon, N.; Bobacka, J.; Viik, J. Bioimpedance Sensor Array for Long-Term Monitoring of Wound Healing from Beneath the Primary Dressings and Controlled Formation of H2O2 Using Low-Intensity Direct Current. Sensors 2019, 19, 2505, doi: 10.3390/s19112505.

Author Response

Manuscript ID: sensors-2802619

Title: Multi-Modal Spectroscopic Assessment of Skin Hydration

Authors: Iman M Gidado, Ifeabunike I Nwokoye, Iasonas F Triantis, Meha Qassem and Panicos A Kyriacou*

The authors would like to thank the editor and reviewers for their valuable suggestions and comments. We addressed the comments and amended the manuscript accordingly. Changes in the manuscript have been highlighted below in blue. Please find below the detailed responses to the reviewers’ comments.

Reviewer 3:

The paper deals with the assessment of skin hydration by using and, concurrently, comparing optical and electrical methods. Specifically, a custom optical solution was compared against the spectrophotometer and the designed bioimpedance measurement solution against the corneometer. In the paper, a multimodal approach is designed, and experimentation is performed on both: porcine and human skin. There are important and interesting results presented in the paper like, among others, the incline in the measured voltage at the optical sensor and bioimpedance measurement unit and the finding related to the measured impedance values after the treatment of the forearm with water. The strength of the paper is revealed in two-stage experimentation (ex- and in-vivo) and multimodal measurement design, supported by gold-standard devices.

However, the reviewer has several concerns/comments/questions:

The novelty of the paper is weakly phrased. What is the innovation that the paper delivers that advances the current knowledge?

  • This paper outlines the testing of custom-designed optical and electrical-based sensing devices, assessing their precision and reliability individually, and evaluating their combined impact. Both ex vivo and in vivo investigations were carried out in order to explore and test the effect of multi-modal measurements and the relationship with one another in the specific case of skin water content. This allowed us to investigate both of the main sensing modalities that can potentially eliminate the aforementioned common corneometer errors (i) and (ii). The multimodal sensing approach, i.e., the use of both electrical and optical sensing modalities can allow for an increased sensitivity and validity in bodily measurements by providing a multi-layer approach of assessment. Furthermore, the in vivo case study ascertained the sensors' accuracy when applied to human skin and allows us to consider the effect of various environmental and individual affecting factors. Analysis of both methodologies facilitated comprehensive insights into the efficacy and the comparative assessment of the individual techniques highlighted their potential complementarity in enhancing the precision of skin hydration measurement.

The ethics statement is fully missing in the drawing, unless measurement experiments are performed, in addition to the porcine skin, on the human forearm. The reviewer also suggests a sentence concerning the Helsinki Declaration.

  • Thank you for your comment. This has now been addressed and the following statements have been stated in the manuscript:
  • “Ethical approval for both porcine skin and human forearm measurement experiments was obtained through a review process conducted by City, University of London Ethics Committee. Informed consent was obtained from the human participant, emphasizing their voluntary participation, confidentiality, and the ethical treatment of data. The study adheres to the highest ethical standards, prioritizing the welfare of both human subjects and animals involved in the research. Furthermore, this study adheres to the principles outlined in the Helsinki Declaration, emphasizing the importance of ethical conduct in research involving human subjects.”

The reason for choosing 1 kHz, 5 kHz, and 10 kHz for excitation signal frequencies in bioimpedance measurements remains unclear. Please add a short explanation or consideration background.

  • Thank you for your comment. In this experiment, the selection of frequencies was influenced by the bandwidth constraints of the electrodes, specifically the limitations of the conductivity sensor. While most corneometer takes measurements at reasonably higher frequencies and up to 1MHZ, it is noteworthy that studies have also explored hydration measurements at lower frequencies, including those below 10 kHz .Therefore, the chosen frequency range in this study aligns with existing literature, taking into account both the practical considerations imposed by electrode capabilities and the broader spectrum of frequencies explored in related research. The manuscript has now been updated to reflect this.

Would be interesting to know the exact type of the used moisturizer and wax to make the experiment repeatable. Can these be named?

  • Thank you for your comment. The results gained from moisturizer and wax have been removed from the in vivo experimental results, leaving dry and water conditions. This is to focus solely on the measurement of the water change on human skin.

The properties of the utilized measuring bioimpedance electrodes (LFT conductivity sensor) need more detailed analysis. It has been claimed that in bipolar bioimpedance measurement, the penetration depth is approx. half of the distance between the electrodes [1]. What is the distance in the case of the used LFT conductivity sensor and how does skin hydration theoretically contribute to the measured bioimpedance values?

  • Thank you for your comment. The conductivity electrodes spacing used in this experiment are I-I 3mm (current injection electrodes), V-V=2mm (voltage measurement electrodes). According to literature the mean epidermal thickness of skin ranges from 76.9 ± 26.2 to 267.4 ± 120.6 µm while the mean dermal thickness ranged from 2115 ± 946.4 to 5888 ± 2422.3 µm. The current injection electrodes were spaced 3mm apart, whereas the voltage measurement electrodes were positioned with a 2mm separation. Consequently, the distance from the centre of the current injection electrodes to their nearest point is approximately 1.5mm, representing an approximate depth of current penetration. Given the average skin thickness of less than 6mm, this configuration implies that the measured boundary voltage resulting from the injected current predominantly originated from the skin top layer. This has now been observed in the manuscript.

Lines 74-75: Bioimpedance hydration sensing relates to the skin's dielectric properties, like permittivity or conductivity, which vary with different water content levels.

In the reviewer’s opinion, the sentence is misleading. The content of dielectrics in the skin remains unchanged while the hydration may change. Typically, the liquid in the skin is conductive (sweat), as originates from the deeper skin layers – affecting the magnitude of the impedance. The dielectrics, like the lipid layer in the stratum corneum, are nonconductive, affecting the phase angle. As a result, the outcome is highly frequency dependent.

  • Thank you for your comments. Dry stratum corneum is a dielectric medium. However, when the stratum corneum is hydrated, a significant change in its dielectrical properties occurs. Within the subcutaneous layer of the skin lie sweat glands that generate sweat, constituting electrolytes. The ionic composition of these electrolytes plays a crucial role in influencing both the skin's conductivity and its dielectric properties. This results in lower skin capacitance compared to non-ionized water in the target measured area. A fundamental aspect guiding conventional corneometer instruments. However, to avoid misinterpretation, the word “dielectric” has now been replaced by the more inclusive term “electrical”.

Lines 85-87: Unlike electrical-based sensors, optical methodologies can penetrate deeper into the skin layers, beyond the epidermis, and provide optical information on specific skin properties [5], [14].

and

  • Thank you for your comment. This has now been changed in the manuscript to: “Optical methodologies have the ability to penetrate deeper into the skin layers, beyond the epidermis, providing more extensive optical information on specific skin properties.”

Lines 577-580: In addition to this, it is suggested that bioimpedance sensors are primarily constrained to assessing the top surface of the skin due to their limited penetration depth, which is a result of the high impedance of deeper skin layers preventing accurate readings beyond the surface.

The reviewer stays in the position that these sentences must be extended to cover the topic. The penetration depth in the case of electrical bioimpedance measurements depends on the dimensions/shape of the electrodes and the gap between the electrodes. This applies predominantly in the case of bipolar measurements, while in the case of tetrapolar system, the term transconductance is and the exact current density distribution is unknown in an object. An excellent reference is the primary textbook in the field of electrical bioimpedance: “Bioimpedance and Bioelectricity Basics” from Grimnes and Martinsen [2] – e.g., starting from chapter 6.4.6 Bipolar Measuring Depth. The impedance of the skin in low frequencies is dominated by stratum corneum [2]. With the increasing frequency, the contribution of lower layers (dermis, hypodermis) expectedly starts to grow. At the same time, when SC becomes moistened (sweat), its conductivity increases.

  • Thank you for your comment. We acknowledge that this could be better explained. The manuscript has now been addressed to reflect this in great detail.

Line 132: The final wearable can be seen in the figure below.

Please refer explicitly to the figure with its number!

  • Thank for your comment. This has been changed to include the specific figure number.

Lines 191-193: On the right side of the sample, was the bioimpedance sensing device, held in place with a small weight to prevent movement of the lightweight sensor.

By applying weight on the bioimpedance sensor that is applied on the skin surface, another phenomenon appears compression of the skin [3]. This has been shown to influence the measured bioimpedance of the skin. Could the applied pressure on the bioimpedance electrode affect the measurement results in the performed study? 

  • Thank you for your comment. According to literature, in bioimpedance measurements, the application of pressure has been shown to reduce electrode-skin interface impedance, particularly in the case of dry electrodes. In this experiment, a consistent pressure of 200 grams was applied uniformly to the porcine specimen throughout. This approach effectively eliminated the potential for variations of results due applied pressure when trying to initiate a good contact during measurements. This is similar to the spring in the corneometer probe which ensures uniform pressure on the skin during measurement. Manuscript has also been adjusted to reflect this.

Figure 12. The title of the figure must be improved to become self-explanatory. Currently, it does not indicate if the presented results are acquired during ex vivo or in vivo experiments.

  • Thank you for your comment. The figure caption has been changed to “Figure 12. Raw voltage output from optical sensor over 6-hour experimental time for selected LED wavelengths with total incremental increases during ex vivo desorption experiment”.

Lines 221-222: The study involved measurements taken on the forearms of the participants under four conditions.

This sentence raises the following questions. How many participants were recruited? How many measurements were performed and are the results the means of some amount of data or a single measurement?

  • Thank you for your comment. The indicative case study involved mean averaged measurements taken from both forearms of a single participant under four conditions. This has now also been stated in the manuscript.

Line 299: “measurements (denoted as C), indicating the influence of electrode/electrolyte interface”

The reviewer can’t find such (denoted as C) from Fig 14 or 15.

  • Thank you for your comment. This has now been corrected and changed to “denoted as T”.

Figure 18 is unclear. What does the “Number” in the x-axis mean? What is this “Corneometer value” in the y-axis? The reviewer assumes that the plot on the left depicts the results before and the plot on the right after the desorption experiment. Would be polite to add A and B on the figure itself.

  • Thank you for your comment. (a) and (b) have now been added to the sub-figures. The caption has been changed to “Figure 18. Corneometer output readings for porcine skin sample before and after ex vivo desorption experiment (Number = Number of Measurements, Corneometer Value = Hydration Index). (a) Hydration Index of Porcine Skin Before Desorption Test, (b) Hydration Index of Porcine Skin After Desorption Test”.

Figure 20. The numbers in the y-axis are not defined – the title of the y-axis is needed. Are the bioimpedance measurement results shown, gained by the bi- or tetrapolar method?

  • Thank you for your comment. The y-axis title with units has been added to the figure.
  • For in vivo measurements, the bioimpedance sensor is in tetrapolar configuration. This has now been stated in manuscript.

Lines 349-350: In the case of the bioimpedance sensor, the voltage output exhibited a sharp increase subsequent to the application of moisturizer.

It is well known that moisturizers, depending on their exact type exhibit large differences in resulting moisturizing effect and interpersonal variations (refer e.g., to [4]). In [4] even cases where the same moisturizer in the case of one person gave a large increase in skin hydration and in the case of the second person no effect was detected, are described. Thus, it is quite hard to generalize, especially as the number of the used subjects and repeated in vivo experimentation is now described in the paper. Can you address this issue?

  • Thank you for your comment. The subsequent drop in voltage when water was added reflects an increase in hydration level, and thus this conforms to the high hydration level index obtained with corneometer when water was added. The manuscript has been rephrased and adjusted to reflect this in great detail.

Lines 350-353: However, the introduction of water yielded an unforeseen decline in the voltage output, contradicting initial expectations. This unexpected drop in hydration measurement, following water application, contrasts with the response observed with the Corneometer device.

The reviewer has a few questions regarding this result: The Corneometer device operates in the MHz region: 0.9-1.2 MHz – reflecting the capacitive properties – capacitive connection to the skin. In bioimpedance measurement, the range of 1 – 10 kHz is much lower. What was the settling time, after which the bioimpedance measurements with the electrodes (which were dry), were performed? Can it be that the skin was not soaked with water yet and the settling time in the case of the dry electrode has been defined to be 15 minutes or more [2] – chapter 4.2.6 Human Skin and Keratinized Tissue.

  • Thank you for your comment. We appreciate your attention to detail and the opportunity to enhance the accuracy of our work. We acknowledge that this statement was inadequately written. A revised version has been made on the manuscript to address this concern. Also, the selection of frequencies was influenced by the bandwidth constraints of the conductivity sensor, however the chosen frequencies remained in line with literature as skin hydration with bioimpedance has also been performed at these frequencies. The manuscript has now been made to reflect this with corresponding references.

Figure 22: the title and x-axis are not visible.

Figure 23: the x-axis of the plots is not visible.

  • Thank you for your comment. The titles and x-axis for both figures have been made larger.

Lines 455-456: Figure 23(a) displays the magnitude of the voltage change (V) for both the optical and bioimpedance results in response to each unit in gravimetric decline.

Can this method be explained and justified more in the paper? How the voltage change at the output of the bioimpedance measurement unit is comparable with the voltage change at the output of the optical sensor? What is the phenomenon (e.g., physiological), that this method relies on, as these methods rely on different sensing technologies?

  • Thank you for your comment. Figure 22(a) displays the magnitude of the voltage change (V) for both the optical and bioimpedance results in response to each unit in gravimetric decline. This shows a greater magnitude of voltage change per gravimetric unit in the bioimpedance over the optical measurements. In addition, Figure 23 displays the box plot outputs for the Kruskal-Wallis statistical analysis for the 3 measurement techniques both experiments, which presents the spread and variance of each technique. This has now been stated in the manuscript.
  • Figure 22(b) has been removed.

Lines 505-507: However, bipolar measurements are more sensitive to variations in electrode-skin impedance and may be less accurate than tetrapolar measurements, especially with smaller skin areas.

Did the authors consider monopolar configuration? Quasi-monopolar configuration is used in skin impedance measurements – for wound healing, for example in targeting a small skin area [5].

  • Thank you for your comment. Major topologies according literature for bioimpedance are mainly bipolar and tetrapolar. Tetrapolar configuration are widely adapted for most bioimpedance sensing due to their ability to minimize the electrode/electrolyte interface. The manuscript has been adjusted to reflect this.

Lines 511-513: In the latter, one pair of electrodes injects current, while another pair measures voltage, eliminating the effects of the electrode-electrolyte interface and enhancing measurement accuracy.

The reviewer would challenge this claim about the elimination of the effects of the electrode-electrolyte interface in the case of tetrapolar measurements. Polarization is a keyword (as practical electrodes are always non-ideal and polarizable) – if the electrode is shifted relative to the skin, the equilibrium state will emerge again, possibly affecting the measured absolute values. When the electrodes are attached to human skin, unintentional movements appear.

  • Thank you for your comment. While it is important to recognize that electrode movements relative to the skin will create a new state of equilibrium, which may be disrupted by polarization, however it is important to note that this study specifically focuses on measurements conducted under steady-state conditions, minimizing the impact of such disruptions.
  • In a bipolar configuration, the injected current traverses through interface impedance, composed of a parallel combination of charge transfer resistance and double-layer capacitance, and in series with electrolyte resistance on its path to the target tissue site. The drawback arises from using the same electrode for both current injection and voltage measurement, leading to erroneous inclusion of voltage drops due to electrode/electrolyte interface impedance in the total measured impedance.
  • In tetrapolar measurements, this issue is alleviated. The current injection and voltage measurement electrodes are distinct, minimizing the impact of interface impedance. This is achieved by measuring the resultant boundary voltages, arising from tissue impedance, via a different pair of electrodes. High input impedance instrumentation ensures a negligible net current flow in the measuring electrodes, allowing for accurate measurement of voltage drops at the target tissue site. Consequently, only the voltage drop at the tissue site is considered and significantly contributes to the total impedance, enhancing the accuracy of bioimpedance measurements.

Lines 602-603: On the other hand, bioimpedance sensors are primarily designed to measure electrical impedance at the stratum corneum.

The reviewer thinks that this sentence needs to be extended and specified. Bioimpedance measurement applications cover a wide variety of target properties and areas of the human body – impedance cardiography, impedance pneumography, body composition, etc., where the interesting area is located inside the body (deeper than the stratum corneum).

  • Thank you for your comment. We acknowledge your comments here and have now adjusted the manuscript to correct this.

References:

[1] P. Aberg, I. Nicander, J. Hansson, P. Geladi, U. Holmgren and S. Ollmar, "Skin cancer identification using multifrequency electrical impedance-a potential screening tool," in IEEE Trans Biomed. Eng., vol. 51, no. 12, pp. 2097-2102, Dec. 2004, doi: 10.1109/TBME.2004.836523.

[2] Ø. G. Martinsen, S. Grimnes, Bioimpedance and Bioelectricity Basics, 3rd ed. Academic Press, Aug 2014

[3] A. Albulbul and A. D. C. Chan, "Electrode-skin impedance changes due to an externally applied force," 2012 IEEE Int. Symp. on Med. Meas. Appl. Proc., Budapest, Hungary, 2012, pp. 1-4, doi: 10.1109/MeMeA.2012.6226628.

[4] Ø. G. Martinsen, S. Grimnes, “Long-Term Effect of Some Skin Moisturizers,” The Open Dermatology Journal, 2008, vol. 2, pp. 87-89, doi: 10.2174/1874372200802010087.

[5] Kekonen, A.; Bergelin, M.; Johansson, M.; Kumar Joon, N.; Bobacka, J.; Viik, J. Bioimpedance Sensor Array for Long-Term Monitoring of Wound Healing from Beneath the Primary Dressings and Controlled Formation of H2O2 Using Low-Intensity Direct Current. Sensors 2019, 19, 2505, doi: 10.3390/s19112505.

  • All suggested references have been included in the manuscript

Round 2

Reviewer 1 Report

Comments and Suggestions for Authors

Thank you for addressing the comments and queries raised in my previous review. I appreciate the effort you have put into revising the manuscript. However, after reviewing the revised manuscript, I find that while the responses to the queries have been adequately addressed, there are still concerns regarding the quality of the figures and the presentation of results.

I strongly recommend that you carefully review and revise the figures to ensure they are visually clear and accurately represent the data, as well as properly aligned in the manuscript. Moreover, consider restructuring the presentation of results to provide a more coherent and organized flow of information, making it easier for readers to grasp the key findings of your study.

Author Response

Reviewer’s Comment

I strongly recommend that you carefully review and revise the figures to ensure they are visually clear and accurately represent the data, as well as properly aligned in the manuscript. Moreover, consider restructuring the presentation of results to provide a more coherent and organized flow of information, making it easier for readers to grasp the key findings of your study.

Response

We sincerely thank the reviewer with this constructive and impactful to the manuscript comment. We have now diligently gone through all the figures and have optimised the quality of the figures, providing more clarity and legibility. We hope that the changes made will suffice, however we remain at your disposal.